# InceptionV3-LSTM: A Deep Learning Net for the Intelligent Prediction of Rapeseed Harvest Time

Shaojie Han [1], Jianxiao Liu [2], Guangsheng Zhou [3], Yechen Jin [3], Moran Zhang [1] and Shengyong Xu [1,*]

1    Key Laboratory of Agricultural Equipment for the Middle and Lower Reaches of the Yangtze River, College of Engineering, Ministry of Agriculture, Huazhong Agricultural University, Wuhan 430070, China
2    College of Informatics, Huazhong Agricultural University, Wuhan 430070, China
3    College of Plant Science & Technology, Huazhong Agricultural University, Wuhan 430070, China
*    Correspondence: xsy@mail.hzau.edu.cn; Tel.: +86-134-7629-3548

**Abstract:** Timely harvest can effectively guarantee the yield and quality of rapeseed. In order to change the artificial experience model in the monitoring of rapeseed harvest period, an intelligent prediction method of harvest period based on deep learning network was proposed. Three varieties of field rapeseed in the harvest period were divided into 15 plots, and mobile phones were used to capture images of silique and stalk and manually measure the yield. The daily yield was divided into three grades of more than 90%, 70–90%, and less than 70%, according to the proportion of the maximum yield of varieties. The high-dimensional features of rapeseed canopy images were extracted using CNN networks in the HSV space that were significantly related to the maturity of the rapeseed, and the seven color features of rapeseed stalks were screened using random forests in the three color-spaces of RGB/HSV/YCbCr to form a canopy-stalk joint feature as input to the subsequent classifier. Considering that the rapeseed ripening process is a continuous time series, the LSTM network was used to establish the rapeseed yield classification prediction model. The experimental results showed that Inception v3 of the five CNN networks has the highest prediction accuracy. The recognition rate was 91% when only canopy image features were used, and the recognition rate using canopy-stalk combined features reached 96%. This method can accurately predict the yield level of rapeseed in the mature stage by only using a mobile phone to take a color image, and it is expected to become an intelligent tool for rapeseed production.

**Keywords:** production estimation; deep learning; rapeseed



## 1. Introduction

As an essential oil crop, rapeseed has become the main source of edible vegetable oil [1]. The harvest period has a significant impact on the quality and yield of rapeseed. A suitable harvest period can be utilized to guide the accurate management, decision-making and marketing of rapeseed [2]. The method of rapeseed harvest is divided into two types: artificial harvesting and mechanized harvesting [3]. Manual harvesting means that harvesting after ripening, drying, threshing, cleaning, and additional steps are completed manually, with low harvesting efficiency and high production costs. Mechanized harvesting of rapeseed includes segmented harvesting and combined harvesting. The operation process of segmented harvest is similar to that of artificial harvest, in which rapeseed is cut down, dried and threshed in the early stage of yellow ripening. Combined harvesting is a combined operation mode of harvesting, threshing, and cleaning rapeseed with a combine harvester, greatly improving the harvesting efficiency [4]. As of 2022, the proportion of manual and machine harvest of rapeseed in China will account for about 50%, respectively. No matter what kind of harvest method, early or late harvesting will have a huge influence on the yield and quality of rapeseed. At present, harvest time is mainly judged by manually observing the color changes of the pods and stalks, which are not only subjective, but also

difficult to replicate. In order to guarantee the quality and yield of the harvest, an intelligent method is urgently needed to determine the best harvest time for rapeseed.

In recent years, the rapid development of computer vision and machine learning has provided a new way to monitor crop growth. At present, hyperspectral analysis technology is widely used to predict crop maturity. For example, Zhao J. [5] and others used the visible/near-infrared spectra of apples with different maturities, extracted spectral feature variables using random forests, and established a fast non-destructive discrimination model for apple harvest periods using extreme learning machine (ELM) and support vector regression (SVR) classification models. Xie Z. and others screened six wavelength combinations by taking hyperspectral images of round leaf spinach, based on the genetic algorithm of grouping elite genetic strategy, and then established a recognition model for spinach freshness based on depth learning technology [6]. Jing Z. [7] and others used UAV to take soybean multispectral images, and predicted soybean maturity time from 130 features in five segments of multispectral images using the partial least squares regression method, and revealed the practicability of multispectral images in predicting soybean maturity. Xu X. and others developed a wheat yield monitoring method by using UAVs remote sensing hyperspectral images and field growth data of winter wheat for many years. In the booting stage, flowering stage and filling stage, R2 values were 0.55, 0.64, and 0.66 [8]. Garcia-Martinez etc. analyzed different multispectral and red-green-blue (RGB) vegetation indexes and digital estimates of vegetation coverage and density, and estimated corn yield using a neural network model [9]. Hyperspectral analysis technology has the advantages of high modeling accuracy but its equipment is expensive and cumbersome to use. It is difficult to promote the application of this technology for rapeseed production, which is dominated by small and medium-sized farms.

In recent years, with the progress of image sensor technology, more and more researchers use RGB images taken by smart phones or UAVs to quickly, and nondestructively, diagnose crops. Laura Z. etc. used the method of multi-camera combinations to capture rapeseed images to analyze and estimated rapeseed yield parameters based on a neural network from single vine images combined with whole rapeseed images [10]. Trevisan Rodrigo etc. used UAV images and developed a complementary convolution neural network to predict the maturity of soybean. The root means square error on the verification set was less than two days [11]. Ji Y. etc. used UAV images to monitor plant height during the whole growth period of broad beans and estimated the yield of broad beans using multiple time point data of plant heights based on a machine learning algorithm. R2 reached 0.98 [12]. Ortenzi L. etc. used UAVs to collect RGB images of olive tree canopies and realized the early yield estimation of olives through the extracted canopy radius. The error of the predicted output was less than 18%. This study provided a digital method for the efficient management of olive plantations [13]. Fathipoor H. established a partial least squares regression (PLSR) model to predict the forage yield by analyzing the plant height and vegetation index obtained based on the RGB image of UAV during the forage growth stage, and R2 reached 0.85 [14]. To sum up, the monitoring method using RGB images has significant low-cost advantages and is easy to carry. However, RGB images are vulnerable to light and have few image features, so it requires professional feature extraction methods and complex machine learning methods to establish accurate prediction models.

Simple and easy intelligent decision-making solutions for harvesting are urgently needed in rapeseed production. The method of combining image feature extraction with machine learning is expected to solve this problem. Most of the existing image feature extraction methods are based on manual extraction, taking the texture, color and their combined features of the image as the input features of machine learning [15]. The artificial extraction method is limited by the influence of lighting conditions, camera sensors, plant surface characteristics and varieties, and it is difficult to establish a stable feature library. The rapidly developing convolutional neural network (CNN) can automatically learn and extract the most representative features from data in the training process with classification as the goal, gradually replacing the artificial feature extraction method [16,17]. However,

the ripening process of rapeseed in the pod stage changes continuously. The image characteristics of rapeseed in the mature stage change gradually with the ripening process. The stalk gradually turns yellow green from near the root to white, while the pod gradually turns yellow and gray from green. Although the CNN can extract complex and effective features, it lacks the continuity of time and is not suitable for such a long dynamic cycle task as rapeseed maturity monitoring. Long short-term memory (LSTM) has been superior in dynamic systalk analysis in many fields [18,19]. CNN has significant advantages in exploring more spatial features, while LSTM has the ability to reveal phenological features. The integration of the advantages of these two types of algorithms will further improve the flexibility and reliability of prediction models based on multi-source production data [20]. For example, SUN and others put forward a soybean yield prediction model based on convolution neural network-long short-term memory network (CNN-LSTM). Taking weather data, surface temperature and ground reflectivity as input data, and taking historical soybean yield data as labels, they combined the data and converted them into tensors based on histograms for model training. The results of the model are better than those of CNN or LSTM models alone. However, this method only takes objective climate data as input, and does not involve the characteristics of soybean growth process [7].

In order to realize the intelligent prediction of rapeseed harvest period, this paper proposed a deep learning network based on CNN-LSTM to analyze the field images of rapeseed in the fruit period to establish a yield classification prediction model. First, the CNN network was used to automatically extract the features of rapeseed canopy HSV color space image with significant features, and the color features of rapeseed stalks screened by the comprehensive random forest method were used to form the rapeseed canopy stalk multi-source feature input, so as to fully exploit the mature rapeseed features. Finally, LSTM was used to deal with the timing problem of rapeseed ripening process, and the experiment showed that the accuracy of yield classification prediction of rapeseed reached 96%. With the presented method, only RGB images taken by mobile phones or drones can be used to accurately determine whether rapeseed has reached the optimal harvest period. It is an intelligent method for rapeseed production.

## 2. Materials and Methods

### 2.1. Experimental Materials and Data Acquisition

The rapeseed varieties were Zhongshuang 6, Dadi 55 and Huayouza 62, which will be planted in Huanggang Modern Agricultural Science and Technology Park, Huanggang City, Hubei Province in 2020 and harvested in May 2021. The experimental field was a rice field rotation, and nitrogen fertilizer was applied at the ratio of 5:2:3 (base, seedling and moss fertilizer). The $P_2O_5$ and $K_2O$ applied in different treatments were 150 kg/ha. All phosphate fertilizers were base-applied, with 1/2 potassium fertilizer as base fertilizer and 1/2 Brassica fertilizer. The method of seedling raising, and transplanting was adopted, sowing on September 24 and transplanting on October 28. Density: 450,000 plants/ha. The width of the irrigation ditch between two adjacent plots was 0.2 m. A 1m wide protective plot was built around the whole site to reduce the edge effect. The experimental data were collected continuously from 1 May to 15 May 2021. A total of 80 plants of each variety were sampled every day. The experimental data collection mainly included three links: image shooting, rapeseed harvesting, threshing and yield measurement.

### 2.1.1. The Experimental Images Acquisition

For the rapeseed harvested on that day, rear cameras such as the iPhone 8P and Xiaomi 10 were used to take images of its canopy and stalk (automatic mode, image resolution of 12 million pixels). Shooting height was about 0.3 m from the canopy and 0.2 m from the stalk. The mobile phone camera was placed to shoot 10–20 images at the canopy and stalk at 17:00–19:00 every day, without direct sunlight. A total of 820 images of three varieties were collected, as shown in Table 1. Specific conditions of note in the shooting environment included: (1) avoiding rainy days and the day following rain, and avoiding

strong reflections caused by direct sunlight on sunny days; and (2) when taking canopy images, attempting to take only images of the pod, and attempting to only take images of stalks when taking images of stalks.

**Table 1.** Image data of rapeseed in mature stage.

| Varieties | Rapeseed Canopy | Rapeseed Stalk | Summation |
|---|---|---|---|
| Zhongshuang 6 | 167 | 107 | 274 |
| Dadi 55 | 155 | 104 | 259 |
| Huayouza 62 | 177 | 110 | 287 |
| Summation | 499 | 321 | 820 |

2.1.2. The Yield Production Acquisition

After the image shooting was completed, 60 rapeseed plants were harvested manually. The daily harvested rapeseed plants were recorded and numbered, and then dried. After drying, the rapeseed met the national recovery standard (moisture content of less than 8%), and manual threshing and weighing were carried out to obtain the daily yield data. Figure 1 shows the curve of daily yield data of the three varieties.

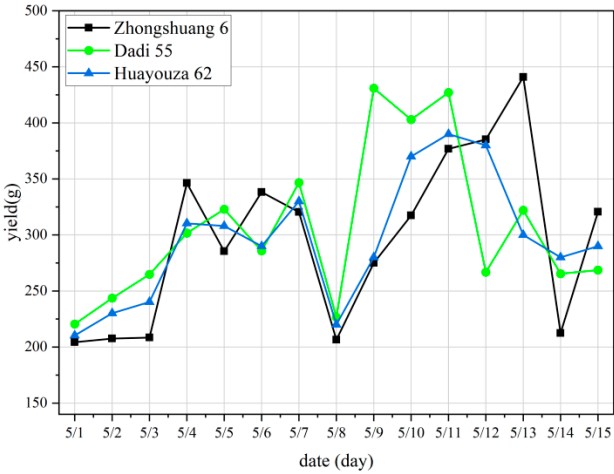

**Figure 1.** Daily yield change of rapeseed over 15 days.

The planting density of the experimental field was 450,000 plants/hm$^2$. We sampled 60 rapeseed plants each time. The production data we acquired was converted into the customary yield data according to Formula (1):

$$C_y = 7.5 \times W_y \ \ \text{Kg/hm}^2 \tag{1}$$

where *Cy* is the customary yield data, *Wy* is the daily yield data we acquired.

*2.2. The Classification Processing of Yield Data*

Nutrient deficiency, soil, climate, cultivation methods, etc. may lead to reductions in yield in varying degrees [21]. The maximum theoretical yield of a rapeseed variety is obtained in the experimental field by a breeding expert, which is only an ideal reference index to establish the yield level. In our manuscript, the maximum yield was provided by the rapeseed research team of the School of Plant Science and Technology in Huazhong Agricultural University. The maturity process of rapeseed can be divided into three stages as green maturity (the pods are green), yellow maturity (10% of the pods begin to turn yellow) and full maturity (most of the fruit pods turn yellow) [22]. According to the suggestions of rapeseed planting experts, the yield range is divided into 90–100%, 70–90% and <70%, which are consistent with the actual agriculture production. According to the

proportion of effective yield and maximum theoretical yield, the problem of absolute yield prediction is transformed into the problem of yield classification prediction. The effective yield is divided into three yield levels, namely, 90–100% yield is first class, 70–90% yield is second class, and less than 70% yield is third class.

Table 2 shows the pretreatment results of yield data of three varieties for 15 consecutive days. The table describes the yield level corresponding to the yield of each variety during the 15-day monitoring. It can be observed from Table 2 that the yield data of 5/8 have an abnormally low value (the rain in the day of 5/8 caused the loss of pod and affected the yield data of 5/9). Even if this abnormal data is not taken into account, the actual yield of the three varieties in the monitoring period is not a linear trend from low to high and then to low. The timing of the highest yield occurrence for the three breeds was also inconsistent. It is difficult to establish a high precision prediction model for the data with insignificant laws using traditional machine learning modeling methods.

**Table 2.** Classification of rapeseed yield data.

| Varieties | First Class | Second Class | Third Class |
|---|---|---|---|
| Zhongshuang 6 | 5/11, 12, 13 | 5/4, 6, 7, 10 | 5/1, 2, 3, 5, 8, 9, 14, 15 |
| Dadi 55 | 5/9, 10, 11 | 5/4, 7, 5, 13 | 5/1, 2, 3, 6, 8, 12, 14, 15 |
| Huayouza 62 | 5/10, 11, 12 | 5/4, 5, 7, 13 | 5/1, 2, 3, 8, 9, 13, 14, 15 |

### 2.3. Image Datasets Enhancement

To improve the generalization ability of the training model, increase the robustness of the model, and at the same time expand the image database of rapeseed in the mature stage, the collected experimental images were enhanced. By randomly rotating the original image to a certain angle, translating a certain distance, scaling, and adding Gaussian noise to these features that do not affect image classification, the purpose of enhancing data was achieved. The fixed enhancement data of canopy and stalk images of three varieties were 250 respectively, that is, 500 images for each variety, 1500 images in total, for subsequent network model training and testing.

### 2.4. Image Segmentation and Feature Extraction of Rapeseed Field Image

#### 2.4.1. Rapeseed Field Image Segmentation Using U-Net Network

In order to eliminate the interference of soil, weeds and other backgrounds on the prediction model, it was necessary to segment the pod from the complex field background for subsequent processing. The deep learning method has been widely used in the field of image segmentation and is also widely used in image segmentation in agriculture [23]. The U-Net network adopts a U-shaped symmetrical structure [24]. This method splices features together in the Channel dimension to form thicker features. In addition, U-Net performs well on small data sets, and has a good recognition effect on subtle edges. Therefore, U-Net is suitable for extraction of slender targets such as rapeseed pod and rapeseed stalk. There are three parts in the U-net. Labeled images and the original images were put into the first four layers. In Downsampling part, the images were resized to four different sizes and high-dimensional feature maps were extracted using convolution. In Contact part, the feature maps of each layer were used to link different modules. Descriptions of the contact modules were shown in the below chart of Figure 2b. In Upsampling part, the convolved high-dimensional feature map was pooled and superposed with the feature map obtained from each layer, and finally connected to the full connection layer for pixel discrimination. A total of 750 canopy images and 750 stalk images in the experimental image set were divided into training set, verification set, and validation set according to 60%, 20% and 20%. The specific network structure and training process of training with U-Net network are shown in Figure 2.

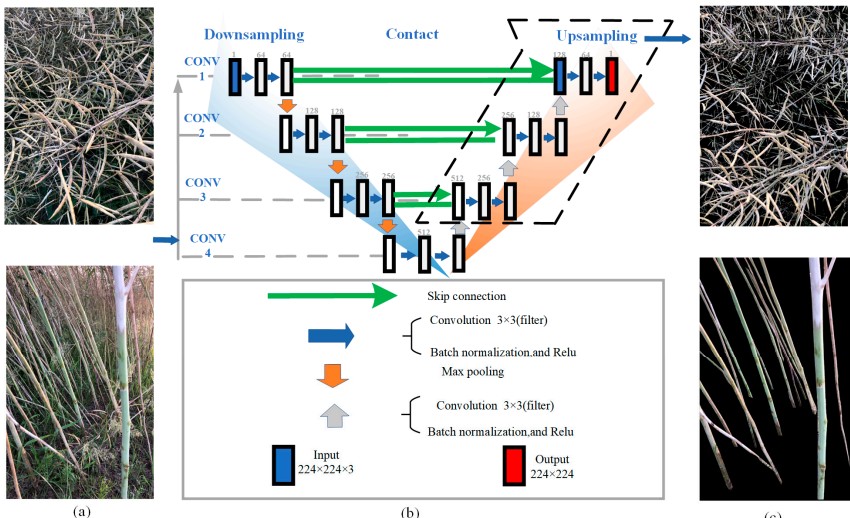

**Figure 2.** U-Net of rapeseed canopy and stalk images. (**a**) Original images of rapeseed canopy and stalk; (**b**) The chart above is the structure of U-Net, the chart below is explanations for the Contact; (**c**) Segmentation results with the U-Net. Note: Downsampling is used to filter features of small effect and information redundancy, by which key information retained. Contact is adopted to merge and enhance the image feature layer. Upsampling enlarges the feature map and restores it to the original size image, in which the segmentation features will display.Conv1~Conv4: Convolution 1 layer to Convolution 4 layer.

### 2.4.2. Silique Image Features Extraction Based on the CNN

The color change of the silique layer is closely related to the maturity. So, accurate extraction of the effective features of the canopy is crucial to the accuracy of subsequent classification and recognition. Compared with RGB color space, the HSV model separates chroma, saturation and brightness, and reduces the influence of light on the characteristics of the silique image. As showed in Figure 3, RGB images of silique in different stages of maturity and its H channel of HSV model are shown. Compared with the RGB image, H channel has a very significant feature change, and the color of mature silique changes from full green to yellow green, and to full yellow and black. Therefore, the H component of HSV color space was used to establish the best harvest time prediction model.

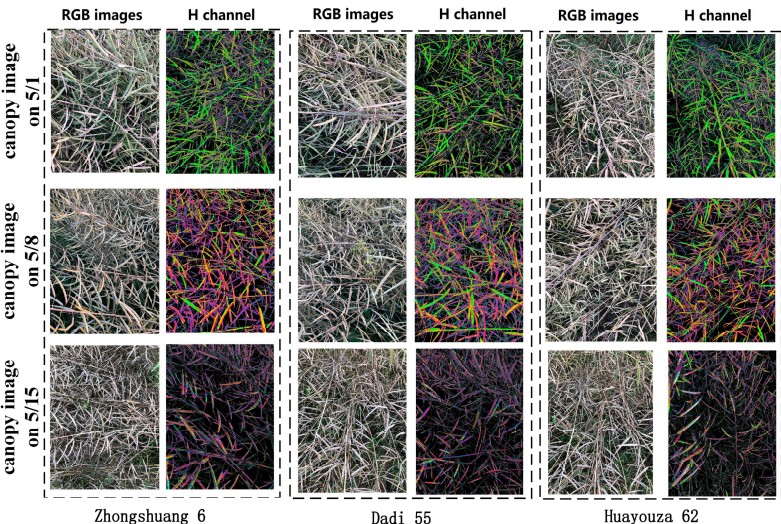

**Figure 3.** Images color changes during the mature period.

To better excavate the spatial, color, and texture characteristics of the rapeseed canopy, the rapeseed canopy images taken for 15 consecutive days were extracted for high-dimensional features using the CNN module. First, the H component of the image is pre-trained. Using $m$ $n \times n$ convolution kernels, the $X_{i,j}$ (the elements in the $i$ th row and $j$ th column of the image) pixels of the image were convolved. After convolution, the value of each pixel in the feature diagram was calculated as showed in Equation (2).

$$a_{i,j} = f\left(\sum_{m=0}^{2}\sum_{n=0}^{2} w_{m,n} x_{i+m.j+n} + w_b\right) \tag{2}$$

In Equation (2), $W_{m,n}$ represents the $n$ th column weight in row m in the convolution kernel, $W_b$ represents the bias term, $a_{i,j}$ represents the $i$ th row $j$ th element in the feature graph, and $f$ indicates the activation function.

After convolution, a total of m feature maps were obtained by average pooling the feature maps, that is, taking the average value within the $n \times n$ range of the feature map to further reduce the image while retaining important information. In the pooling of the last layer, the results were arranged into a row of vectors $x_i$ to form a fully connected layer. Finally, extracted high-dimensional features were input into the LSTM for training to establish a classification model.

### 2.4.3. Stalk Image Features Extraction Based on the Random Forest

In agricultural production, the yellow-green-ratio of stalks is typically used as an important indicator to judge whether rapeseed is mature. In order to more comprehensively and accurately describe the changes of rapeseed stalk color during rapeseed ripening, 31 color features of the stalks were extracted from RGB, HSV, YCbCr and other three spaces [25]. Among them, seven color features of vegetation index based on RGB spatial model, such as EXR, were adopted, as shown in Table 3. For HSV and YCbCr color space, the mean, standard deviation, skewness, and peak values of the three channels were selected, respectively.

**Table 3.** Color features of the RGB color model.

| Color Space | Color Characteristics | Abbreviation | Computational Equation |
|---|---|---|---|
| RGB | Excess red | ExR | $1.4r - g$ |
| | Normalized differential vegetation index | NDVI | $(G-R)/(G+R)$ |
| | Excess green | ExG | $2g - r - b$ |
| | Excess green minus excess red | ExGR | $ExG - ExR$ |
| | Color index of vegetation extraction | CIVE | $0.441r - 0.811g + 0.385b + 18.78745$ |
| | Vegetation | VEG | $G/R^{0.667}B^{0.333}$ |
| | Combination of vegetation index | COM | $0.25ExG + 0.3ExGR + 0.33CIVE + 0.12VEG$ |

Note: R-image R channel mean; G-image G channel mean; B-image B channel mean; r-image R channel normalized value; g-image G channel normalized value; b-image B channel normalized value.

In order to solve the problem of too many original features and redundancy, stalk color features were screened to improve the effectiveness of features. Random forests can effectively screen the color features of stalks by internal sorting of features [26]. A training sample set was formed by taking part of the sample from 31 kinds of eigenvalues in a random and put-back manner, and repeated $N$ times to form $N$ training sample sets and $N$ decision trees. The features that are not drawn as tests were used to calculate the error rate of the decision tree model prediction, which became the out-of-pocket data error ($err_{ooB_1}^q$). Noise interference was randomly added to the features not extracted and the out of bag data error again was calculated again, which was recorded as $err_{ooB_2}^q$. For $N$ trees, the importance of feature $q$ was calculated as shown in Equation (3) [27].

$$E(q) = \sum_{n=1}^{N}\left(err_{OOB_1}^q - err_{OOB_2}^q\right)/N \tag{3}$$

where, $E(q)$ is the feature importance; $N$ is the number of classification decision trees; $q$ is the current calculation feature; If the accuracy of out of bag data decreases significantly

(that is, $err_{ooB_2}^q$ increases). After adding random noise, this indicated that feature $q$ has a great impact on the prediction results of samples, which further indicated that it is of high importance. The color characteristics of the top six rapeseed stalks obtained from random forests were: the average value of channel A, the average value of channel B, the average value of channel CB, ExR, NDI, and the average value of channel Y. These six features are the key image features of the stalk.

### 2.5. Prediction Model of Yield Level Based on CNN-LSTM

LSTM is a special Recurrent Neural Network (RNN), which is mainly used to solve the problem of gradient disappearance and gradient explosion during long sequence training. Compared with the general neural network, it can deal with the data of sequence changes. For the continuous time series of rapeseed maturity, the maturity of the previous day must have a great relationship with the maturity of the next day and will affect the maturity of the next day. In addition, the maturity of rapeseed in the field is uneven, and the effective yield data fluctuates in a small range. If only image features are used without considering the relationship between images, it will lead to a wrong prediction. Transfer of cell state h in LSTM structure exactly describes the relationship between images. For continuously collected image datasets, the use of CNN-LSTM can not only fully mine the feature information carried by a single image itself, but also can fully use the continuity between features to mine the timing information carried between images, which can maximize the accuracy of discrimination. Therefore, this paper used the neural network architecture based on CNN-LSTM to establish the prediction model of rapeseed harvest time.

The processing of recognition using the CNN-LSTM framework is shown in Figure 4. The input of this framework is the silique features using CNN and the stalk features using Random Forest. Initial state is a set of initial parameters for h0 adopted by human set-ting, which will change from zero to other value in the deep learning net training. Yield production was to be classed three levels, so there were 3 classes in the CNN features extraction. There were three classes in the final recognition result too. High dimensional features are extracted from the last full connection layer of CNN and used as one-dimensional vector features of $x_t \times 1$, where $t$ is the number of images in each time step. Each gate cell contains an $m \times n$ matrix, m represents the canopy pod features extracted by CNN and the stalk features obtained by machine learning screening, and n represents all time steps. As a loss function, cross entropy adjusts network parameters based on classification accuracy in the dataset. After the training, the final hidden layer unit state $h_{t-1}$ represents the important feature coding information of each period. The one-dimensional feature vector $h_{t-1}$ is connected to the Full connection layer, and then connected to a *Softmax* layer for classification.

LSTM structural parameters are shown in Table 4. This paper determined hyperparameters based on experimental results from large datasets.

**Table 4.** LSTM network structure parameters.

| Parameters | Specification |
| --- | --- |
| Training (%) | 70 |
| Testing (%) | 30 |
| Feature size | 3 |
| Input gate | Sigmoid |
| Forget gate | Sigmoid |
| Output gate | Sigmoid |
| Hidden layer | Tanh |
| Number of layers | 1 |
| Number of Hidden units | 100 |
| Loss function | Cross-entropy |
| Optimizer | Adam |
| Epoch | 150 |
| Batch size | 8 |

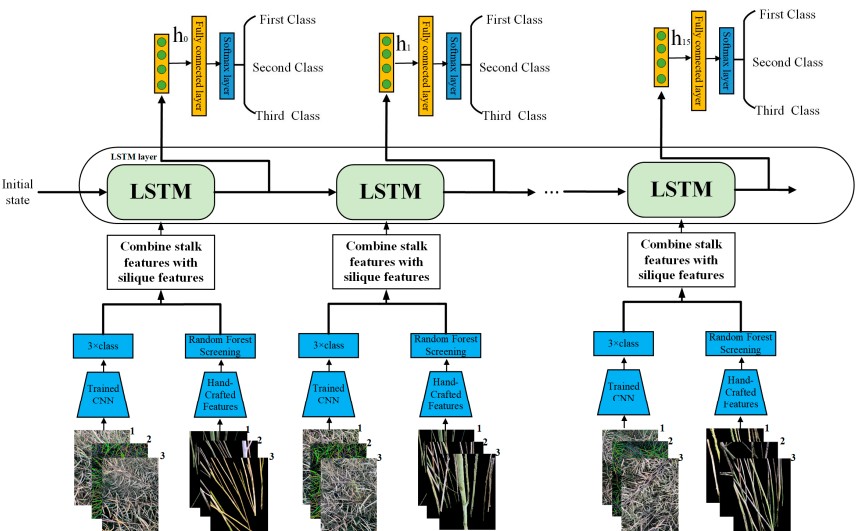

**Figure 4.** Structure of the prediction model using CNN-LSTM.

## 3. Results and Analysis

The development and test platform adopted the deep learning server (Core i5-9300H/ 16G/512G SSD/NVIDIA GeForce GTX1660Ti 8G) and the software framework of Python 3.6, TensorFlow 1.13 and Keras 2.1.1. In order to verify the effectiveness and robustness of the model, 70% of the experimental data were randomly divided into training sets, and the remaining 30% were used for testing. This paper mainly used the accuracy (*Acc*), precision (*Pr*), recall (*Re*) and other indicators to evaluate the performance of the CNN-LSTM model. Formulas (4)–(6) were shown as follows.

$$Acc = \frac{TP + TN}{TP + FP + FN + TN} \tag{4}$$

$$Pr = \frac{TP}{TP + FP} \tag{5}$$

$$Re = \frac{TP}{TP + FN} \tag{6}$$

In the formulas, *TP* (true positives) means the positive class is determined as a positive class, *FP* (false positives) means the negative class is determined as a positive class, *FN* (false negatives) means the positive class is determined as a negative class, and *TN* (true negatives) means the negative class is determined as negative class.

### 3.1. Classification Performance Test of CNN-LSTM Using Silique Features

This experiment was mainly for the verification test of five CNN networks, including VGG16, EfficientNet V1, ResNet18, MobileNet V1 and Inception V3. The classification layer of the network was adjusted to three layers (corresponding to three output levels), and the learning rate was 0.001, the batch size was 8, and the input image size was 224 × 224. One pot coding was used to train and extract image features, and the best network for feature extraction of rapeseed canopy image was selected to fully explore the spatial features of mature rapeseed canopy. The t-SNE dimension reduction method was used to visualize high-dimensional features to a two-dimensional plane. Figure 5 shows the visualization results of five CNN network feature extraction performance. Figure 5a–e are characteristic diagrams of VGG16, EfficientNet V1, ResNet18, MobileNet V1 and Inception V3 respectively. It can be seen from Figure 5 that VGG16, EfficientNet V1 and InceptionV3 networks divide the three types of feature points perfectly, while the other two networks cannot divide the features well. Furthermore, compared with VGG16 and EfficientNet

V1, InceptionV3 is more focused, which shows that InceptionV3 has better robustness and generalization for each class.

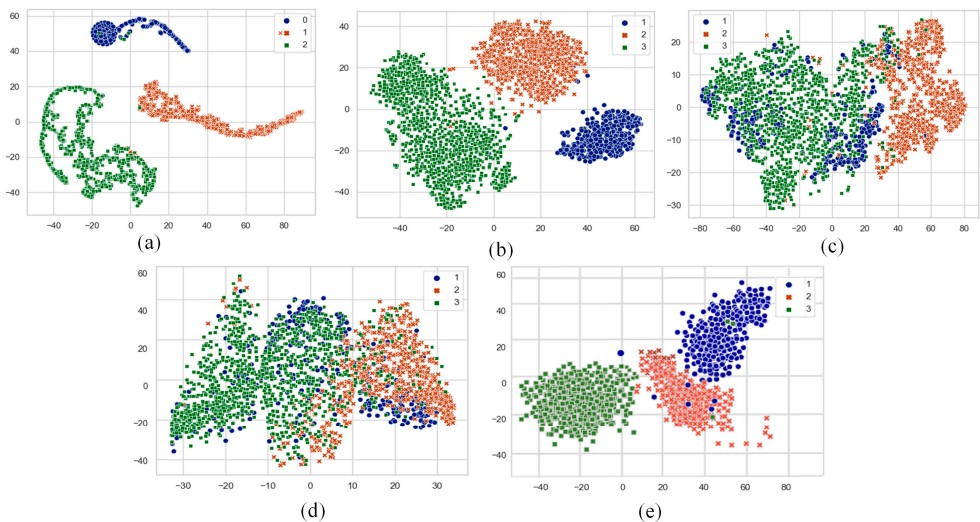

**Figure 5.** Visualization of five CNN networks: (**a**) VGG16; (**b**)Inception V3; (**c**) ResNet18; (**d**) MobileNet; and (**e**) EfficientNet V1.

In addition, this paper evaluated the performance of CNN-LSTM and CNN models, and the results are shown in Table 5. Five CNN networks were used to classify and predict the yield levels of three varieties of rapeseed. Using the first variety, the average recognition rates of VGG16, Inception V3, ResNet18, EfficientNet V1 and MobileNetV1 corresponding to the three varieties were 78%, 83%, 69%, 77%, and 53%, respectively. The CNN network itself has classification and prediction capabilities. Prediction results using CNN alone are shown in the upper part of Table 4. The results are not very satisfactory. Therefore, we did not use CNN network to predict directly but to extract image features. Canopy image features extracted from CNN network were used as input, and LSTM recurrent neural network was further used for training and classification. The recognition accuracy of CNN-LSTM combined network pattern has been greatly improved, and the recognition rate of Inception V3-LSTM combined network in three varieties reached more than 90%. The experimental results show that the prediction accuracy of CNN-LSTM has been significantly improved compared with only using CNN.

**Table 5.** Prediction accuracy of CNN and CNN-LSTM of 3-levels classification.

| Features | Model | First Variety | | | Second Variety | | | Third Variety | | |
|---|---|---|---|---|---|---|---|---|---|---|
| | | Acc (%) | Pr (%) | Re (%) | Acc (%) | Pr (%) | Re (%) | Acc (%) | Pr (%) | Re (%) |
| VGG16 | | 78 ± 1 | 78 ± 1 | 78 ± 1 | 72 ± 1 | 76 ± 1 | 78 ± 2 | 75 ± 1 | 78 ± 1 | 76 ± 1 |
| Inception V3 | | 83 ± 1 | 84 ± 1 | 85 ± 1 | 83 ± 1 | 84 ± 1 | 85 ± 1 | 82 ± 1 | 83 ± 1 | 82 ± 1 |
| Resnet18 | None | 69 ± 2 | 66 ± 2 | 67 ± 2 | 59 ± 2 | 57 ± 2 | 57 ± 2 | 70 ± 1 | 62 ± 2 | 69 ± 2 |
| EfficientNetV1 | | 77 ± 2 | 79 ± 2 | 76 ± 2 | 73 ± 2 | 72 ± 2 | 77 ± 2 | 73 ± 2 | 74 ± 2 | 78 ± 2 |
| MobileNet V1 | | 53 ± 3 | 50 ± 3 | 49 ± 3 | 45 ± 3 | 50 ± 3 | 51 ± 3 | 56 ± 3 | 51 ± 3 | 48 ± 1 |
| VGG16 | | 90 ± 1 | 89 ± 1 | 90 ± 1 | 89 ± 1 | 88 ± 1 | 90 ± 1 | 90 ± 1 | 89 ± 1 | 89 ± 1 |
| Inception V3 | | 91 ± 1 | 93 ± 1 | 91 ± 1 | 94 ± 1 | 92 ± 1 | 92 ± 1 | 94 ± 1 | 91 ± 1 | 92 ± 1 |
| Resnet18 | LSTM | 76 ± 1 | 75 ± 1 | 75 ± 1 | 73 ± 1 | 74 ± 1 | 75 ± 1 | 75 ± 1 | 74 ± 1 | 77 ± 1 |
| EfficientNetV1 | | 89 ± 2 | 88 ± 2 | 90 ± 2 | 91 ± 2 | 87 ± 2 | 89 ± 2 | 91 ± 2 | 90 ± 2 | 90 ± 2 |
| MobileNet V1 | | 63 ± 3 | 63 ± 3 | 63 ± 3 | 61 ± 3 | 59 ± 3 | 62 ± 3 | 63 ± 3 | 58 ± 3 | 63 ± 3 |

Using the best two nets, we supplied the test results of 5-level classification with the presented method and the results are shown in Table 6. The yield range was divided into 90–100%, 80–90%, 70–80%, 60–70% and <60%. The experimental results are shown in the table. Compared with the 3-level classification method, the overall accuracy rate of the 5-level classification method decreased by about 6%. More yield ranges require higher differentiation of image features. However, yield ranges both of 60–70% and 70–80% corresponded to the green maturity period, and the difference of image features is not obvious, which will decrease the recognition rate markedly.

**Table 6.** Test results of 5-level classification of yield production.

| Features | Model | First Variety | | | Second Variety | | | Third Variety | | |
|---|---|---|---|---|---|---|---|---|---|---|
| | | Acc (%) | Pr (%) | Re (%) | Acc (%) | Pr (%) | Re (%) | Acc (%) | Pr (%) | Re (%) |
| VGG16 | LSTM | 83 ± 1 | 83 ± 1 | 84 ± 1 | 83 ± 1 | 83 ± 1 | 83 ± 1 | 83 ± 1 | 83 ± 1 | 85 ± 1 |
| Inception V3 | | 84 ± 1 | 84 ± 1 | 85 ± 1 | 84 ± 1 | 84 ± 1 | 85 ± 1 | 85 ± 1 | 84 ± 1 | 85 ± 1 |

*3.2. Classification Performance Testing Using the Silique-Stalk Dual Feature*

The color characteristic of the stalk is an important basis to judge the maturity of rapeseed in agricultural production. A large number of crop yield estimation studies often only consider the canopy and ignore the stalk. Therefore, in this experiment, the double features formed by the color features of stalks extracted from random forests and the silique features of rapeseed were used as input, and then the yield level was predicted using CNN-LSTM network. The classification results are shown in the blue histogram in Figure 6. The blues are the recognition accuracy of the three varieties using only silique features. The oranges are the recognition accuracy using silique-stalk combined features. It can be seen that, after the addition of stalk features, the classification and recognition rate of rapeseed yield of each network has been improved. The recognition rate of InceptionV3-LSTM network reached more than 96%. The stalk color feature can effectively improve the accuracy of the model and verify the correctness of agricultural production experience.

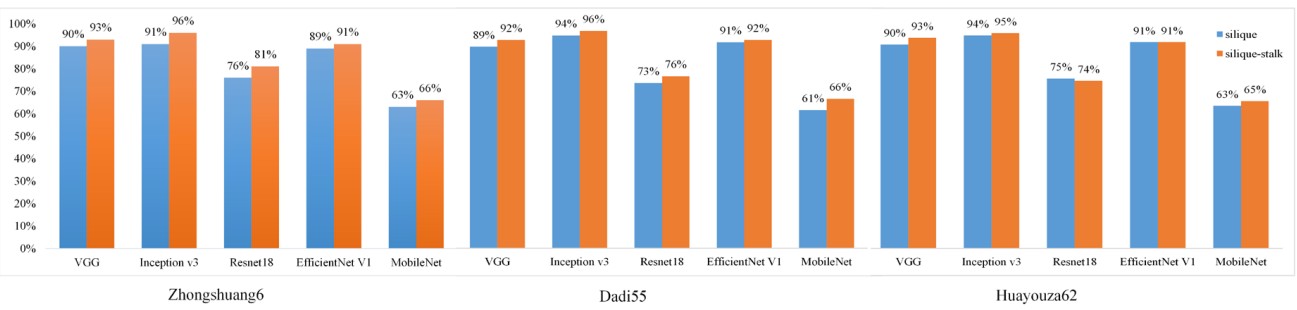

**Figure 6.** Comparison of silique features and silique-stalk combined features of 3-levels classification.

Additionally, the Inception V3-LSTM network was tested with the canopy-stalk combined features and the canopy features respectively. The results are shown in Figure 7. Adding stalk features can improve the prediction accuracy by 2–5% effectively.

Figure 8 shows the confusion matrix of InceptionV3-LSTM model on training set and validation set. In the training set, the recognition rate of the first two levels reaches 100%, and that of the third level also reaches 98%. In the validation set, the recognition rate of the second level and the third level is very high. Compared with the other two levels, the recognition rate of the first level is lower. This is because the maturity dates of the first stage and the second stage are close to each other, resulting in the similarity between the rapeseed image and the second stage and the third stage, which leads to the misjudgment of the image, but the overall recognition rate is 96%.

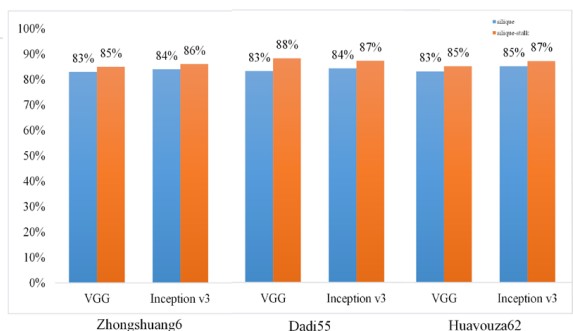

**Figure 7.** Comparison of canopy characteristics and silique-stalk characteristics in the 5-level classification with the Inception v3-LSTM.

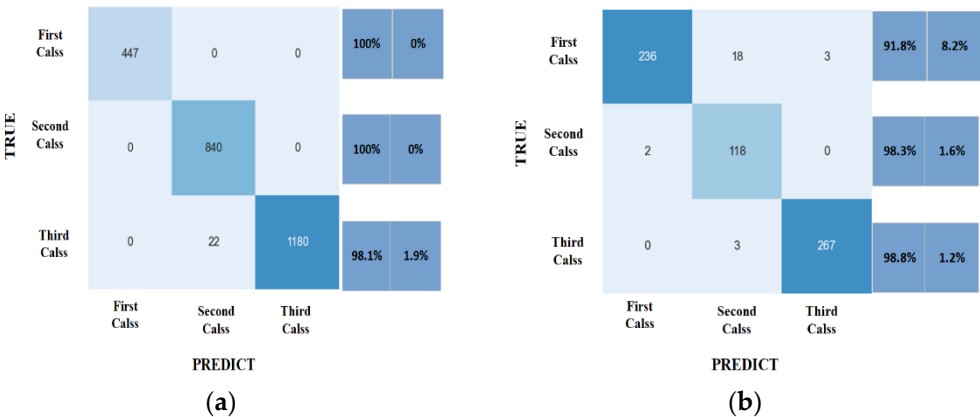

**Figure 8.** Confusion matrix of the Inception V3-LSTM. (**a**) Training set; (**b**) Validation set.

Different datasets have different effects on model accuracy during training. The impact can be reduced by adjusting the proportion of rapeseed image database in mature period. Figure 9 shows the stability of our model when using different training set sizes. Figure 9a,b are the results of training using 100% and 70% datasets, respectively. The axis of Iteration is iterations count. The axis of Accuracy and Loss are the accuracy and loss rate of training process respectively. Although the accuracy fluctuates, it is stable after 200 iterations. This shows the superiority of the model in this paper. Fewer datasets can be used to establish a prediction model.

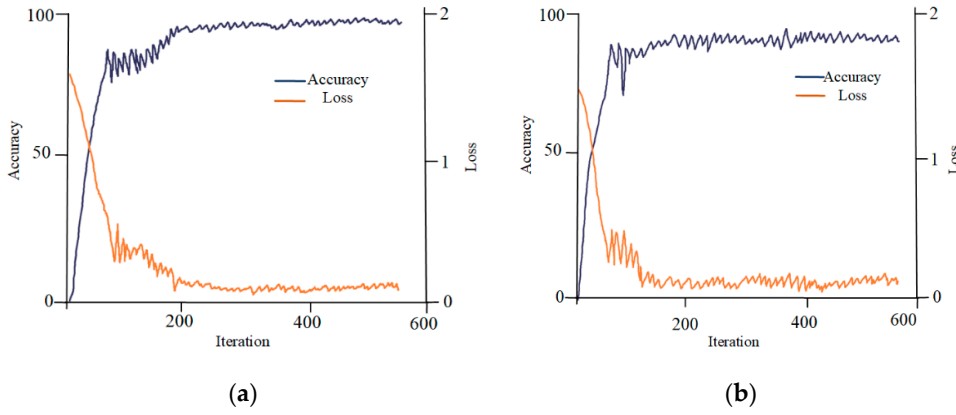

**Figure 9.** Training process of Inception V3-LSTM framework. (**a**) Training process using 100% datasets; (**b**) Training process using 70% datasets.

## 4. Conclusions

In this paper, a prediction model of the best harvest time of rapeseed based on the CNN-LSTM deep learning framework was proposed. Based on rapeseed canopy and stalk image data at pod stage, high-dimensional features of rapeseed canopy image were extracted by CNN deep learning method, and then the color features of rapeseed stalks were screened by using random forests, forming the joint features of rapeseed silique and stalk. Based on the joint characteristics of the above two aspects, a prediction model of Inception V3-LSTM rapeseed yield grade was established by using LSTM recurrent neural network. The experimental results showed that 94% classification accuracy was achieved in the validation set for three rapeseed varieties. Using CNN network to extract image features to replace the traditional artificial feature extraction method can not only greatly reduce the invalid workload, but also help to avoid the impact of many redundant features on the recognition rate of the algorithm and has a good robustness for images with insignificant color features. In addition, referring to the experience of manually judging the maturity of rapeseed, the color feature of the stalk was added, which can improve the classification and recognition rate by 6%. Using Timeline tools of TensorFlow, the algorithm execution time was tested with 12-million-pixel images of rapeseed canopy and stalk. The average result of 20 time was 0.54 s.

We present a precise prediction method for the three varieties of rapeseed. For a new rapeseed variety, the presented method will be adopted for prediction of harvest time through the following steps: (1) Breeding experts provided the maximum theoretical yield data of a new rapeseed, which is divided into three levels according to three intervals of 90–100%, 70–90% and <70%; (2) Image and yield data were collected from the yellow maturity of rapeseed. Canopy and stalk images of rape were taken for 8–12 consecutive days, and 60 rapes were taken to measure yield; (3) The presented method was used to process the collected images and yield data to obtain the yield level prediction model; and (4) Steps (2) and (3) only need to be carried out in the first year, and can be applied in the second year and after. After entering the yellow maturity period, workers can monitor the change of rape yield by taking the canopy and stalk images and process these images. If the prediction results show that the yield level is between 90–100%, harvesting can be carried out. In the case of serious reduction conditions, the prediction result will remain at a low level even at the full maturity period, and can be harvested immediately.

The method proposed in this paper has realized automation for the feature extraction of canopy and stalk. When the prediction model is established for more varieties and larger data, the model can be constructed efficiently. Therefore, the method proposed in this paper has good adaptability and practicality. Furthermore, the data and model in this article will be deployed to the cloud database and open to users for free. Users can use mobile terminals to judge the best harvest time of rapeseed conveniently and quickly, which can effectively improve the intelligence of rapeseed production and reduce yield loss.

**Author Contributions:** Conceptualization, S.X.; methodology, S.X. and S.H.; validation, S.H. and Y.J.; formal analysis, S.X.; investigation, G.Z.; resources, G.Z.; data curation, S.H. and Y.J.; writing—original draft preparation, S.H. and J.L.; writing—review and editing, M.Z. and S.X.; supervision, J.L. and S.X.; funding acquisition, G.Z. All authors have read and agreed to the published version of the manuscript.

**Funding:** This research was funded by the National Key Research and Development Program of China, grant number 2018YFD1000904.

**Data Availability Statement:** The data used to support the findings of this study are available from the corresponding author upon request.

**Conflicts of Interest:** The authors declare that they have no conflict of interest.

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
