# Peer review of "InceptionV3-LSTM: A Deep Learning Net for the Intelligent Prediction of Rapeseed Harvest Time"

_agronomy, doi:10.3390/agronomy12123046_

Round 1
Reviewer 1 Report
In this paper, a prediction model of the best harvest time of oilseed rape based on the CNN-LSTM deep learning framework is proposed. Using CNN network to extract image features to replace the traditional artificial feature extraction method can not only greatly reduce the invalid workload, but also help to avoid the impact of many redundant features on the recognition rate of the algorithm and has a good robustness for images with insignificant color features. In this paper, the specific conditions of the shooting environment should be clarified for the convenience of users.Here are some suggestions for modification:
Materials and Methods:
Line 172: (1) is not an equation. Please recheck the wording on line 171 and line 173.
Line 172: The grading criteria are not clear.
Line 194: The first letter of the title should be capitalized.
Line 260: " equation " should be " Equation ".
Line 156: Tables and figures should have a full stop at the end of their title sentences. Similar problems exist in the following tables titles and figures titles.
Results and analysis:
Line 368: The first line of this paragraph should be indented.
Author Response
Thank you for your letter and the reviewers' comments on our manuscript (ID:agronomy-2013615). Those comments are constructive for revising and improving our paper, as well as the important guiding significance to further research. We have studied the comments carefully and made corrections which we hope meet with approval.

Reviewer 2 Report
Authors have proposed a deep learning net for the intelligent prediction of oilseed rapes harvest time. The CNN and LSTM network are used to establish the oilseed rape yield classification prediction model. The experimental results show that Inception v3 of the five CNN networks has the highest prediction accuracy. This method can accurately predict the yield level of oilseed rape in the mature stage by only using a mobile phone to take a color image, and it is expected to become an intelligent tool for oilseed rape production. I have following problems: 1.The yield data in Figure 2 is the result in a single acquisition, which needs to be converted into the commonly used yield data . 2.In section 2.2 The hierarchical processing of yield data, according to the proportion of effective yield and maximum theoretical yield, the problem of absolute yield prediction is transformed into the problem of yield classification prediction. Is the standard of the classification? Have other works used this standard to do prediction? Maybe authors should be given the corresponding reference work. 3.In the section of Conclusion, authors should further discuss and analyze the advantages and disadvantages of the method. And analyze the reason why their method performing well. In addition, the next step work should be strengthen. 4.The time consuming of the diagnosis method can affects the practical value. Please supplements the execution time data of the algorithm. 5.In the manuscript the yield was classified to three categories. If five or more categories, what ‘s difference in the experimental results? 6.The English level of the whole manuscript should be improved. 7.Some pictures maybe too large, such as Figure 8. Authors should do some improvement.
Author Response

(The authors gave the same response as above.)

Reviewer 3 Report
1. When describing the research content of this paper in line 120, a separate paragraph can be added so that readers can better locate and understand the relevant content of the paper.
2, Line 136, tense is not appropriate, planting and harvesting has been completed, the article is a statement and summary of what has been done, there is no will.
3. The reasons for lines 145 and 146 are the same as in Article 2.
4. Figure 6 should be placed below paragraph 289. First the text, then the picture.
5. The calculation formula of Acc, pr, Re and other evaluation indicators can be listed in line 304, which is more convincing.
6. Table 4, None and LSTM would be better to add a dividing line.
Author Response

(The authors gave the same response as above.)

Reviewer 4 Report
This paper presents a very interesting CNN-LSTM machine learning approach for predict the yield of rapeseed yield. Although the use of CNN-LSTM’s is not a novel on agriculture, namely other plant harvests such as rice and soy, its application to rapeseed is new.
Major comments
1. The results presented by the authors seem to be good, but they are not compared with other prediction methods described in the literature, which is one of my major concerns regarding the manuscript. The authors should compare their results with other methods (even if those method are not based on deep-learning). A direct comparison is probably too difficult and too long to perform, but at least a small discussion addressing this matter should be provided.
2. Another concern are layout issues, inconsistent designations that appear differently on the figures and text, and some important references missing (see below). As for the English language it generally reads well, but the text contains a few glitches.
a) Table and figure revisions:
Figures and tables are not correctly numbered (for instance, there is no Figure 5; there are two Table 4’s). Correct them and check their references in the text.
Figure 3: It is not clear where “Upsampling” is applied to. Also, the U-Net diagram lacks image quality (image format should be vector graphics, if that is not possible, it should have a much greater DPI).
l215-216 – “As shown in Figure 4, RGB images and HSV images of silique with different maturity are shown.” – revise the sentence. “RGB images” and “HSV images” is an inaccurate designation, since the image is the same and the only thing that differs is the color representation. In fig 4, it is not clear what color channels are represented.
Fig. 4 reads “Combine stem features with crown features”. However, “stem” and “crown” are not ever addressed in the text. Please address them in the adequate sections.
Table 4 (the first table 4) – correct “Iuput gate”
The width of Figure 8 is too large to fit in the text margins.
The figures depicting the Confusion Matrices must be regenerated – they contain the “Calss” glitches and class names are not clear… shouldn’t they be the “Excellent”, “Normal” and “Fail” classes mentioned in section 2.2?
Fig. 10-b) Seem to be address the validation set, not the test set.
b) Missing references:
In section 2.4.1, the author use the designation “UNet”, but the official designation is “U-Net”. The reference for the U-Net segmentation neural network is not ever provided:
Ronneberger, O., Fischer, P., Brox, T. (2015). U-Net: Convolutional Networks for Biomedical Image Segmentation. In: Navab, N., Hornegger, J., Wells, W., Frangi, A. (eds) Medical Image Computing and Computer-Assisted Intervention – MICCAI 2015. MICCAI 2015. Lecture Notes in Computer Science(), vol 9351. Springer, Cham. https://doi.org/10.1007/978-3-319-24574-4_28
The information regarding the indexes in table 3 lacks the corresponding references – please provide them on the table or on the subsection 2.4.3’s text.
l271-272 – Provide a reference for LSTM – a book, or the original paper that proposed LSTM:
Sepp Hochreiter, Jürgen Schmidhuber, “Long Short-Term Memory”, Neural Computation, Vol 9(8), November, 1997, DOI: 10.1162/neco.1997.9.8.1735
c) Minor revisions/glitches
l149 – “2.1.1. The experimental images were taken” – the title is strange – please review
l156 – “oilseed rape” is duplicated in table 1’s caption
l194 – check subsetcion 2.4.1 title’s capitalization
l197-198 – check sentence: “The depth learning method has been widely used in the field of image segmentation…”, namely the designation “depth learning” which seems strange to me.
l304 “(pr)” -> “(Pr)”
3. l309-310 – “and the remaining 30% is used for verification.” -> “and the remaining 30% is used for testing.” or “and the remaining 30% is used for validation.”? By reading the text it seemed to me that the authors use a “validation set” instead of a “testing set” (since the “validation set” is the one verified during training – the testing set should be used after training the models).
4. l313 – It is not clear which version of the MobileNet was used. What is the motivation for using AlexNet? Its architecture is somewhat similar to VGG16, which is slightly more recent. On the other hand, why not use more recent CNN models (e.g., EfficientNet, Xception)?
Author Response

(The authors gave the same response as above.)

Round 2
Reviewer 2 Report
The authors have well addressed my concerns.